# A Study of the Relationship between the Dynamic Viscosity and Thermodynamic Properties of Palm Oil, Hydrogenated Palm Oil, Paraffin, and Their Mixtures Enhanced with Copper and Iron Fines

**DOI:** 10.3390/ma17071538

**Published:** 2024-03-28

**Authors:** Agnieszka Dzindziora, Damian Dzienniak, Tomasz Rokita, Jerzy Wojciechowski, Maciej Sułowski, Saltanat Nurkusheva, Michał Bembenek

**Affiliations:** 1Department of Manufacturing Systems, Faculty of Mechanical Engineering and Robotics, AGH University of Krakow, A. Mickiewicza 30, 30-059 Krakow, Poland; dzindziora@agh.edu.pl (A.D.); ddamian@agh.edu.pl (D.D.); 2Departament of Mechinery Engineering and Transport, Faculty of Mechanical Engineering and Robotics, AGH University of Krakow, A. Mickiewicza 30, 30-059 Krakow, Poland; rokitom@agh.edu.pl; 3Department of Power Systems and Environmental Protection Facilities, Faculty of Mechanical Engineering and Robotics, AGH University of Krakow, A. Mickiewicza 30, 30-059 Krakow, Poland; jwojcie@agh.edu.pl; 4Department of Physical Metallurgy and Powder Metallurgy, Faculty of Metals Engineering and Industrial Computer Science, AGH University of Krakow, A. Mickiewicza 30, 30-059 Krakow, Poland; sulek@agh.edu.pl; 5Department of Organization of Transport, Traffic and Transport Operations, L. N. Gumilyov Eurasian National University, Satbaev 2, Astana 010000, Kazakhstan; 6Department of Transport Equipment and Technologies, S. Seifullin Kazakh Agrotechnical Research University, Zhenis 62B, Astana 010011, Kazakhstan

**Keywords:** phase-change materials, PCMs, palm oil, paraffin, phase transition

## Abstract

The article presents the results of phase transition studies in which the following substances and their mixtures were tested: 100% palm oil, 100% paraffin, 100% hydrogenated palm oil, 50% palm oil + 50% paraffin, 50% hydrogenated palm oil + 50% palm oil, 33% hydrogenated palm oil + 33% palm oil + 33% soft paraffin, 20% hydrogenated palm oil + 30% palm oil + 50% soft paraffin, 50% hydrogenated palm oil + 50% palm oil + copper, and 50% hydrogenated palm oil + 50% palm oil + iron. The measurements were carried out on a station for testing phase-change materials (PCMs) designed specifically for the analysis of phase changes. Viscosity values were also determined for the tested materials, and their potential impact on heat accumulation was assessed. The primary goal of the experiment was to determine some key thermodynamic parameters, including transition time, transition heat, specific heat, and dynamic viscosity at 58 °C. A one-way ANOVA test confirmed the statistical significance of minimum transition temperature, maximum transition temperature, and phase transition time, validating the reliability and utility of the results. The melting point, crucial for applications involving phase changes, was identified as an important factor. The careful selection of components allows for the customization of properties tailored to specific applications. A significant result is that the analyzed substances with higher specific heat values tend to have a higher average dynamic viscosity. The Pearson correlation coefficient of 0.82 indicated a strong positive association between the average dynamic viscosity and the heat of fusion of the substances examined. This suggests that changes in the heat of fusion significantly influence alterations in dynamic viscosity. Substances with higher specific heat values tend to exhibit higher average dynamic viscosity, emphasizing the direct impact of composition on viscosity.

## 1. Introduction

Phase-change materials (PCMs) are substances characterized by high heat of fusion that undergo melting and solidification processes at a constant or nearly constant temperature and efficiently absorb or release thermal energy from or to the environment, facilitating effective thermal management [1]. An example of utilizing phase change (e.g., melting and solidification) for storing thermal energy is PCM, whereas hydrogen storage facilities utilize phase changes (e.g., liquefaction and vaporization) for storing chemical energy [2]. The choice of the most efficient PCMs depends on the specific application and requirements regarding temperature, thermal stability, performance, and costs [3]. PCMs can be divided based on chemical composition into organic, inorganic, and eutectic [4].

Organic materials are characterized by stability in melting cycles. The heat of phase change ranges from approximately 200 to 250 kJ/kg [5]. A common issue associated with organic PCMs is the need for large temperature changes to achieve phase transition. Organic materials have low thermal conductivity but high volume expansivity during melting. Examples of organic PCM materials include paraffins, esters, fatty acids, and ionic liquids [6,7].

Inorganic materials most commonly occur as hydrated salts and less frequently as metals. PCMs of this type have a dense structure. Compared with organic materials, inorganic PCMs are more efficient but also much more expensive. The drawback of inorganic materials is their propensity to cause corrosion. Inorganic PCMs also face issues with supercooling, meaning that they do not remain crystalline after phase transition. Examples of inorganic PCMs include hydrated salts [8,9,10].

Eutectic materials are formed as mixtures of different substances. They are most commonly formed from combinations of organic–organic substances, inorganic–inorganic substances, and organic–inorganic substances [11,12].

Table 1 below compares a selection of PCMs based on their properties.

One of the problems discussed in the article is low thermal conductivity [15]. PCMs have relatively low thermal conductivity during the phase transition, and that limits the rate at which thermal energy can be stored or released [16]. Enhancing thermal conductivity without sacrificing the storage capacity is a significant challenge [17,18]. To address the challenge of low thermal conductivity in PCMs, several technological solutions have been adopted [15]. These include the incorporation of thermally conductive additives or fillers, such as carbon nanotubes, graphene, or metal nanoparticles, into the PCM matrix to enhance thermal conductivity [19,20]. Another approach involves the development of composite PCMs, where the PCM is encapsulated or dispersed within a thermally conductive matrix material. Furthermore, advanced manufacturing techniques, such as microencapsulation or nanostructuring, are utilized to control the morphology and distribution of PCM particles, thereby improving overall thermal conductivity [21]. Microencapsulation (about 0.1–1000 μm in diameter) involves covering core material particles with a thin shell, creating microencapsulated phase-change material slurry (mPCMS) [22,23,24,25]. Therefore, the research attempted to dope the samples with conductors, i.e., copper or iron [26].

The aim of this research was to investigate the influence of parameters such as phase transition temperature and dynamic viscosity and to determine parameters such as fusion heat, specific heat, and transition time [27].

Viscosity tests of PCMs are important from the point of view of many applications, such as energy storage and transmission, cooling and heating, as well as in the field of electronics, construction, and other engineering fields [28].

Viscosity affects a material’s ability to flow and move between areas of different phase states [29]. Materials with lower viscosity may move more easily, which may be important in some applications—which was also the aim of the study.

Materials with the appropriate viscosity can enable efficient heat transfer during phase change. The viscosity affects the stability of the material structure during phase change cycles. Materials that are too viscous may experience difficulty maintaining their structure as they repeatedly transition between phase states [30]. Similar studies can be found in the literature regarding PCM substances, but encapsulated and available as aqueous dispersions [30,31,32].

Traditional methodology for measuring thermodynamic properties includes differential scanning calorimetry (DSC) and differential thermal analysis (DTA). These methods are widely employed techniques for assessing thermal properties. They are particularly useful for determining latent heat and typically necessitate only small, uniform samples [15,32]. In this research, a specially designed chamber was used for heating PCM and recording temperature. It is a simple device with which one can test the parameters of the material being studied. Our research used a station that, unlike the traditional one, uses Peltier modules (for heating and cooling) and temperature sensors. In order to determine the thermodynamic parameters, it is necessary to perform calculations, which can also be implemented into the program controlling the entire device. Compared with traditional methods of determining specific heat, this is an inexpensive way that can be easily reproduced. The research on PCMs also includes determining viscosity and phase transition time and searching for relationships between the determined thermodynamic parameters for palm oil, hydrogenated palm oil, paraffin, and their mixtures. Despite the commonality of the studied materials, similar studies for such mixtures have not been found. Knowledge of liquid viscosity plays a pivotal role in the design and formulation of various materials, such as paints, adhesives, coatings, or cooling fluids. Proper viscosity selection allows for obtaining optimal application properties, such as adhesion, fluidity, or diffusion ability. Knowledge of PCM viscosity is important in the design process of these systems because it can affect their performance and stability. For example, viscosity can affect the rate of heat spreading within the PCM and the rate of phase transition, which may be important for the efficiency of the thermal regulation system. Determining parameters such as viscosity or specific heat for the test substances can be important from the application level. In some cases, a lower viscosity may be preferable, allowing faster heat transfer and phase transformations. However, in other cases, a higher viscosity may be preferable, providing material stability and control over flow and thermal behavior. Despite the widespread use of the materials used for testing, there are not many studies showing, for example, properties for mixtures.

## 2. Materials and Methods

### 2.1. Characteristics of the Tested Substances

The samples were provided by Korona Candles, Wieluń, Poland, which mass-produces candles from these substances, among others, e.g., palm oil, hydrogenated palm oil, and soft paraffin. PCM substances exhibit different phase transformation temperatures depending on the composition. The research problem was the temperature range of phase transformation and the ability of the substance to accumulate the heat of the tested samples. Nine samples were tested. The composition of the samples was varied, thanks to which three groups of tested samples can be distinguished. The first group comprised “pure substances”, the second contained mixtures of the first group, whereas the third included composites based on sample 3 with metallic additives.

#### 2.1.1. Palm Oil

The melting point of palm oil is in the range of 33–39 °C. The kinematic viscosity at a temperature of 58 °C is at a level of 35 mPa·s (which is a measured value; unfortunately, the manufacturer does not have such data). The substance is used on a large scale in the food and cosmetic industries [33]. All the available properties of the palm oil are listed in Table 2.

#### 2.1.2. Hydrogenated Palm Oil

Hydrogenated palm oil is stable, oxidizes more slowly, and is more resistant to high temperatures. The melting point is in the range of 55–58 °C [34]. The kinematic viscosity at a temperature of 58 °C is at a level of 50 mPa·s (again, this is a measured value since the manufacturer does not provide such data). Those and the remaining available properties of the hydrogenated palm oil are included in Table 3.

#### 2.1.3. Soft Paraffin

The phase transition temperature range for soft paraffin is 45–50 °C [35]. The kinematic viscosity at a temperature of 58 °C is at a level of 49.5 mPa·s (a measured value). All the available properties of the soft paraffin are shown in Table 4.

#### 2.1.4. Added Substances

The problem with substances such as paraffin or palm oil is their low thermal conductivity. This parameter is characteristic of a given substance in a specific state of aggregation. Various materials can be used to increase thermal conductivity, e.g., iron filings or copper [21,36].

#### 2.1.5. Copper

The substance exhibits a purity of at least 99.5% and is distinguished by a fine powder with grains smaller than 0.063 mm. The thermal conductivity coefficient for commercial copper is 372.16 W/mK. During the research, copper was used to increase the thermal conductivity of the tested PCM [37]. The choice of a fine fraction was made to increase the surface area of the copper particles, facilitating better dispersion within the PCM.

#### 2.1.6. Iron Fines

Iron fines intended for didactic research were used in the study—fine powder, main fraction 45–150 µm. The value of the thermal conductivity coefficient is about 54.66 W/mK. The purpose of the filings was to increase the thermal conductivity of the tested sample [37].

Table 5 shows the substances that were tested.

### 2.2. Testing Station

The test stand (Figure 1a) consisted of three areas: a functional system, a regulating system, and software.

The functional system was designed to create a heating–cooling chamber, allowing the conditions necessary for the phase transition of the examined substance. The functional system included Peltier modules, fans, radiators, a container with a PCM, and housing. Thanks to this solution, it is possible to use Peltier modules as a cooling and heating source [38].

The regulation system included elements such as a converter, relays, Arduino, displays, resistors, and temperature sensors. These devices were responsible for powering and controlling the entire functional system. The system was controlled using an Arduino microcontroller. The software, Arduino IDE 2.3.2, allows one to write, compile, and upload code to the Arduino board. The electrical diagram is shown in Figure 1b.

The program was responsible for turning on and off the heating sources and regulating their temperature by controlling temperature sensor readings. This function allowed for recording measurements to obtain the characteristics of the tested material. The displays in the system showed the temperature of the lower and upper heat sources.

### 2.3. Preparation of the Stand and Samples for Testing

#### 2.3.1. Sample Preparation

Approximately 10 g of material was prepared for each test. The material was placed in a glass beaker attached to a Peltier module using a thermal paste to enhance heat flow.

#### 2.3.2. Test Stand Configuration

The test stand included a heating chamber made of Peltier cells, where phase change occurred [39]. The chamber temperature ranged from 60 to 70 °C. The research station had sensors for measuring temperature on the upper and lower Peltier modules and in the beaker with the material. The temperature of the modules was controlled via the program in the range of 60–70 °C. The housing was constructed of plexiglass to limit heat exchange with the environment.

#### 2.3.3. Temperature Measurement and Phase Transformation Curve

Temperature measurements were recorded by a sensor, generating a phase transformation curve (heating curve). The PCM was heated until it reached about 60–70 °C. Natural cooling was used. This methodology allowed the system temperature to decrease gradually and naturally.

### 2.4. Viscosity Measurement

Seven types of samples were tested: 100% palm oil, 100% paraffin, 100% hydrogenated palm oil, 50% hydrogenated palm oil + 50% paraffin, 50% hydrogenated palm oil + 50% palm oil, 33% hydrogenated palm oil + 33% palm oil + 33% soft paraffin, and 20% hydrogenated palm oil + 30% palm oil + 50% soft paraffin. The dynamic viscosity test was performed using the Brookfield method at a speed of 100 RPM, spindle no. 1, and at a temperature of 58 °C.

Brookfield viscosity measurement involves the use of a device known as a Brookfield viscometer [31]. The device measures the resistance that a substance offers to the rotational motion of a rotor inside a sample of liquid. Viscosity measurements can be conducted under various conditions, such as different rotor speeds, temperatures, and rotor geometries, allowing for customization of the measurement to a specific application [40]. The measurement temperature was selected in the liquid phase of the substance to prevent issues with substance deposition (subcooling) on the spindle [41].

In summary, the Brookfield viscosity measurement methodology was selected for its adaptability, dynamic viscosity capabilities, temperature suitability, widespread usage, and the ability to fine-tune parameters for the specific nature of the samples being tested in the study.

### 2.5. Statistical Tools

In order to reject the default null hypothesis, which would suggest that the composition of the substances had no bearing on the observable output parameters, a one-way ANOVA (analysis of variance) test was performed. For each type of sample, 10 measurements were taken, in which such parameters as the transition temperature, phase transition time, and process time were recorded (although the latter had been discarded, since it was a rather arbitrary value). Those parameters, excluding the processing time, were bound together as the dependent variables for the purposes of the statistical test. Since the transition temperature usually had minimum and maximum values, it was split into two columns that corresponded to those boundary values.

### 2.6. Calculation Algorithm Used in the Analysis

Based on the measurements, it was possible to determine the heat of transformation for the solid state and the specific heat for palm oil, hardened palm oil, soft paraffin, and mixtures of these substances. An algorithm was developed to derive these parameters.

#### 2.6.1. Algorithm for Calculating Specific Heat Capacity

Determination of the heat necessary to heat the beaker [42,43]:(1)$$Q_{z}=m_{s}\cdot c_{s}\cdot ∆T$$
where
$Q_{z}$—heat the beaker [J];$m_{s}$—mass of the glass [kg];$c_{s}$—specific heat capacity of the glass $\left[\frac{kJ}{kg\cdot K}\right]$;∆T—temperature change [°C].

Calculation of the heating heat of the system:(2)$$Q_{el}={\dot{Q}}_{el}∆t=(U\cdot I+Q_{Peltier})∆t$$
where
${\dot{Q}}_{el}$—heat flux generated by power supplies and cell [W];∆t—heating time of the substance [s];U—voltage [A];I—current intensity [V];$Q_{Peltier}$—heat flux of the Peltier module [W].

Determination of the specific heat for the solid state of the PCM [43]:(3)$$c_{w}=\frac{Q_{PCM}}{m\cdot ∆T}=\frac{Q_{el}-Q_{Z}}{m\cdot ∆T}$$
where
$c_{w}$—specific heat for the solid state of the PCM $\left[\frac{kJ}{kg\cdot K}\right]$;$Q_{PCM}$—heat of the PCM [J];$Q_{el}$—heat generated by heating the system [J];$m_{s}$—mass of the substance [kg];∆T—temperature change [°C];$Q_{z}$—heat the beaker [J].

#### 2.6.2. Algorithm for Calculating Fusion Heat Capacity

Calculation of the heat flux generated by the power supplies and the module [44]:(4)$${\dot{Q}}_{el}=U\cdot I+Q_{Peltier}$$

Determination of the heat of the tested PCM (heat balance):(5)$$Q_{el}=Q_{PCM}$$

Calculation of the heat of transformation for the PCM:(6)$$Q_{fusion}=\frac{Q_{PCM}}{m_{s}}$$

## 3. Results and Discussion

It can be concluded from Table 6 that the composition affects the phase transition range. Palm oil, which is a “pure substance”, had the lowest temperature. Hydrogenated palm oil exhibited the broadest spectrum of phase-change temperatures, with a total range of 10 °C, spanning from 45 to 55 °C. This may have been due to the addition of hardeners to pure palm oil. The tested samples were characterized by “sharp temperature peaks”, which shows the poor thermal conductivity of the tested substances. The problem was eliminated by the addition of copper to 50% palm oil with 50% hydrogenated palm oil, which shortened the time to reach the maximum temperature and extended the cooling time. The disadvantage of composites is the lack of a noticeable phase transition boundary, but with proper observation, it is possible to determine this parameter. A similar experimental recording of parameters has been reported in the literature for PCM substances via another test station [45].

### 3.1. Eutectics of the Tested Substances

Pure substances and their mixtures of various concentrations were tested. The ranges of phase transformations were obtained, on the basis of which the eutectic diagram of the tested samples was created (Figure 2). A strict dependence of the phase transition temperature on the composition was obtained [46] as follows:For the substance containing 100% palm oil, the transformation temperature was 35 °C;For the system with a concentration of 75% palm oil and 25% hydrogenated palm oil, the transformation temperature was 39–40 °C;For the system with a concentration of 50% palm oil and 50% hydrogenated palm oil, the transformation temperature was 40–44 °C;For the system with a concentration of 25% palm oil and 75% hydrogenated palm oil, the transformation temperature was 42–45 °C;For 100% hardened palm oil, the transformation temperature was 50 °C [47,48,49,50].

### 3.2. Viscosity Measurement

Based on the test results (Figure 3), the following can be concluded:Soft paraffin has a relatively low average viscosity of 27.86 mPa·s compared with the other substances on the chart. This is beneficial in terms of the fluidity and movement of the substance;Palm oil has a higher average viscosity of 45.94 mPa·s than soft paraffin. This means it may be less liquid and thicker;Hydrogenated palm oil has an even higher viscosity of 54.24 mPa·s than palm oil. It is a substance with a higher density and less liquid;The mixture of 50% palm oil and 50% soft paraffin appears to have an average viscosity of 32.56 mPa·s between pure palm oil and soft paraffin;The mixture of 50% hydrogenated palm oil and 50% palm oil appears to have an average viscosity of 52.28 mPa·s, similar to hydrogenated palm oil alone;The mixture of hydrogenated palm oil (33%), palm oil (33%), and soft paraffin (33%) appears to have a moderate viscosity of 38 mPa·s;Another blend, hydrogenated palm oil (20%) + palm oil (30%) + soft paraffin (50%), appears to have a moderately low viscosity of 32.06, perhaps dominated by soft paraffin.

The viscosity of a substance can be a key factor in many applications, such as transportation, industry, and manufacturing. The choice of specific formulation can be adjusted depending on the viscosity expectations for a given application.

Mixtures of different substances can lead to the desired viscosity tailored to specific application needs [49].

As a solid melts into a liquid, the viscosity usually decreases. The melting process loosens the crystal structure, which in turn facilitates the flow of molecules. As a result, the viscosity of the liquid (molten form) is usually lower than the viscosity of the solid. Checking the viscosity level for mixtures can result in the possibility of determining in which substance particles flow most easily and most difficultly. Thanks to the viscosity test, it is possible to determine which substance exhibits the most favorable flow characteristics. Consequently, among the tested substances, it is possible to identify the one in which heat transfer will occur most effectively.

For substances based on soft paraffin, palm oil, and a mixture of substances, there is a linear correlation. It can be seen that palm oil has the lowest melting point but the highest average dynamic viscosity.

Dynamic viscosity in heat storage materials refers to the ability of a substance to resist deformation due to dynamic loading, such as fluid flow in channels, pipes, or heat conduction systems. PCMs are materials that can store and release heat through phase transitions, such as during melting and solidification [23].

Dynamic viscosity (sometimes also called kinematic viscosity) is a measure of a material’s resistance to flow. In the case of PCMs, dynamic viscosity can affect the flow rate of the substance during phase transition and the efficiency of storing and releasing thermal energy [41].

Average viscosity and average melting point data analysis results (Figure 4) are as follows:Soft paraffin appears to have a moderate viscosity of 27.86 mPa·s and a higher melting point of 49.5 °C;Palm oil has a relatively high viscosity of 45.94 mPa·s and a lower melting point of 35 °C;Hydrogenated palm oil is characterized by a high viscosity of 54.24 mPa·s and a moderately high melting point of 50 °C;The mixture of 50% palm oil and 50% soft paraffin appears to have a moderate viscosity of 32.56 and a moderately high melting point of 45 °C;A blend of 50% hydrogenated palm oil and 50% palm oil appears to have a high viscosity of 52.28 and a moderately low melting point of 42 °C;The mixture containing the three ingredients—hydrogenated palm oil (33%), palm oil (33%), and soft paraffin (33%)—has a moderate viscosity of 38 mPa·s and a melting point of 38.5 °C;Another blend—hydrogenated palm oil (20%) + palm oil (30%) + soft paraffin (50%)—appears to have a moderately low viscosity of 32.06 mPa·s and a moderately high melting point of 40 °C.

The choice of specific composition depends on the intended use. Blends can combine various characteristics, such as viscosity and melting point, to adapt to specific application conditions.

Blends with soft paraffin appear to lower viscosity, which may be beneficial in some applications.

Melting point is also an important factor, especially in the context of applications where phase changes are crucial. Appropriate selection of ingredients can enable the properties to be tailored to a specific application [50].

The Pearson coefficient of 0.87 for the tested substances indicates a strong positive correlation between the two variables. A correlation level of −1 to 1 means that the closer the absolute value of the Pearson coefficient to 1, the stronger the correlation [51].

In the case of a coefficient of 0.87, it can be concluded that the variables usually move together in one direction, and the correlation between them is strong. In the context of analyzing data on the viscosity and phase transition time of substances or their mixtures, these values may suggest that there is a tendency for a correlation between these two characteristics. In other words, a change in one variable often goes hand in hand with a change (Figure 5) in another variable.

There is a certain tendency for an inverse correlation between viscosity and phase transformation time. Substances with higher viscosity tend to have a longer phase transition time, and substances with lower viscosity tend to have a shorter phase transition time.

Blends such as “palm oil 50% + soft paraffin 50%” and “hydrogenated palm oil 33% + palm oil 33% + soft paraffin 33%” show mild changes in viscosity (32.56 mPa·s and 38 mPa·s, respectively) at maintaining a short phase transformation time. This suggests that moderate viscosity can be achieved without significantly affecting the phase transition time.

Blends such as “hydrogenated palm oil 20% + palm oil 30% + soft paraffin 50%” demonstrate balanced properties, combining moderate viscosity with a relatively short phase transformation time. Such combinations may be beneficial in applications where both features are important.

Blends with hydrogenated palm oil (e.g., “hydrogenated palm oil 50% + palm oil 50%”) seem to have a higher viscosity and a longer phase transformation time. This suggests that this substance may be more suitable for applications requiring phase stability over time.

The addition of soft paraffin (e.g., “palm oil 50% + soft paraffin 50%”) may reduce the viscosity of the mixture, which may be beneficial in some applications without significantly affecting the phase transformation time.

In summary, the selection of the appropriate formulation depends on the specific application requirements, where substance combinations can be tailored to achieve the desired viscosity properties and phase transformation time. The choice of blend composition can be tailored to the specific requirements of a given application, balancing viscosity and phase transformation time.

### 3.3. Statistical Analysis

The results of the statistical ANOVA analysis are presented in Table 7.

As can be seen above, the *p*-value for each of the observed parameters is extremely small (with the F-value being relatively high), so even with a very stringent significance level (e.g., 0.01), it is easy to prove that the results and, therefore, the differences are statistically significant. Additionally, for each of the three responses (minimum transition temperature, maximum transition temperature, phase transition time), Tukey’s honest significance test was performed to compare the possible pairs of the obtained means since the analysis of variance does not reveal much about the relationships between the respective sample types. The confidence level was chosen to be 0.99 (99%), which corresponds to a 0.01 (1%) significance level. The results of the test are visually presented on the plots in Figure 6, Figure 7 and Figure 8 [49].

The width of the confidence intervals for the minimum transition temperature is approximately 2. For the maximum transition temperature, it is slightly above 1, and for the phase transition time, it falls between 7 and 8.

### 3.4. Comparison of Tested Substances

In order to analyze the results in detail, several comparisons were made for the different samples that had been tested. Three samples were compared, and the comparison was presented in Figure 9: 50% hydrogenated palm oil + 50% paraffin + copper, 50% hydrogenated palm oil + 50% paraffin + iron, and sample 50% hydrogenated palm oil + 50% paraffin. The addition of thermal conductors extends the duration of the entire process by a factor of three. The heating time is also extended without a clear phase transition boundary. During the heat transfer from the samples for composites, a narrow phase transition range was obtained [52,53].

### 3.5. Experimental Determination of Parameters

The specific heat capacity and the fusion heat were determined on the basis of research carried out for palm oil, soft paraffin, and hydrogenated palm oil. The experimental specific heat for palm oil is 2 $\frac{kJ}{kg\cdot K}$ and is within the theoretical range. The obtained fusion heat is 117.28 $\frac{kJ}{kg}$ and is slightly different from the theoretical values.

For soft paraffin, the experimental specific heat was 2.87 $\frac{kJ}{kg\cdot K}$ and exceeded the theoretical range. The obtained heat of transformation was 109.95 $\frac{kJ}{kg}$ and differed slightly from the theoretical values. The same parameters were determined for hydrogenated palm oil. They were 2.21 $\frac{kJ}{kg\cdot K}$ and 146.6 $\frac{kJ}{kg}$, respectively; thus, they fell within the range of theoretical values (Table 7). In addition, 20% Hydrogenated palm oil + 30% palm oil + 50% soft paraffin has a relatively low heat of fusion of 99.7 $\frac{kJ}{kg}$ and a specific heat capacity of 1.91 $\frac{kJ}{kg\cdot K}$. The dynamic viscosity is moderate at 32 mPa∙s.

Also, 50% Hydrogenated palm oil and 50% paraffin has an average heat of fusion of 117.28 $\frac{kJ}{kg}$ and a moderate specific heat capacity of 1.93 $\frac{kJ}{kg\cdot K}$. It is characterized by a moderate dynamic viscosity of 32.56 mPa∙s.

Next, 50% hydrogenated palm oil and 50% palm oil has a relatively high heat of fusion of 131.93 $\frac{kJ}{kg}$ (among the substances tested) and a low specific heat capacity of 1.87 $\frac{kJ}{kg\cdot K}$. It is also characterized by a high dynamic viscosity of 52.28 mPa∙s.

However, 33% hydrogenated palm oil + 33% palm oil + 33% soft paraffin has a low heat of fusion of 102.62 $\frac{kJ}{kg}$ and a specific heat capacity of 1.85 $\frac{kJ}{kg\cdot K}$. Dynamic viscosity is moderate at 38 mPa∙s [32].

The obtained results (Figure 10 and Figure 11) prove the accumulating capacity of the tested substances. Mathematical determination of thermodynamic parameters can be used as an approximate determination of the properties of materials for heat accumulation [53,54].

Analyzing the data from Table 8, it can be seen that paraffin has the highest accumulating capacity and also has the lowest dynamic viscosity. For the tested substances, there is a correlation between the heat of fusion and the dynamic viscosity. The higher the melting heat, the higher the viscosity of the tested substances.

The value of the Pearson correlation coefficient is 0.82, indicating a strong positive relationship between the average dynamic viscosity and the heat of fusion of these substances. This means that changes in heat of fusion have a significant impact on changes in dynamic viscosity. The higher the specific heat value, the higher the average dynamic viscosity [59,60].

The heat of fusion is an important factor influencing the dynamic viscosity of these substances. An increase in heat of fusion is usually associated with higher dynamic viscosity, and a Pearson correlation coefficient of 0.82 confirms this correlation. Understanding this is of practical importance, especially in an industrial context where viscosity control can be crucial to production processes and product quality [32].

Analyzing the correlation between these two variables can help improve processes and optimize products, especially in fields such as the chemical, cosmetics, or food industries.

Each sample appears to have unique characteristics regarding heat of fusion and dynamic viscosity. This suggests that different substances or mixtures may exhibit differences in these properties.

## 4. Conclusions

Seven samples with different compositions were tested: soft paraffin, palm oil, hydrogenated palm oil; palm oil (50%) + soft paraffin (50%); hydrogenated palm oil (50%) + palm oil (50%); hydrogenated palm oil (33%) + palm oil (33%) + soft paraffin (33%); hydrogenated palm oil (20%) + palm oil (30%) + soft paraffin (50%). Two additional samples were mixtures with conductors; however, they were not the main focus of the research. The tested samples showed poor thermal conductivity. In order to minimize the problem with the thermal conductivity of the tested substances, two samples were created: mixture 1 and mixture 2. Mixture 1 is sample no. 6 with copper rods, and mixture 2 is sample no. 6 with iron filings. The graphs obtained did not have a clear phase transition temperature. This is due to the heating factor of the system. The process of temperature change itself is dynamic, while the cooling process is prolonged, which proves that the “working” time of the batteries made of the tested composites is extended.

The aim of the experiment was to determine the main thermodynamic parameters, i.e., transition time, transition heat, specific heat, and dynamic viscosity in the liquid state at 58 °C. In some cases, lower viscosity may be more advantageous, allowing faster heat transfer and faster phase transformations, which is what soft paraffins do in our research. However, in other cases, a higher viscosity may be preferred, providing material stability and control over its flow and thermal behavior, examples being substances such as hydrogenated palm oil or 50% hydrogenated palm oil and 50% palm oil.

Among the investigated substances, the highest specific heat is for paraffin. However, the highest heat of transformation was recorded for hydrogenated palm oil, which can be desirable in some PCM applications. Higher specific heat means a greater amount of energy that the PCM material can store per unit mass. Higher heat of transformation means a greater amount of energy required for the phase change of the PCM material. Such parameters can be advantageous in applications where storing a larger amount of energy is crucial.

A one-way ANOVA test for the minimum transition temperature, maximum transition temperature, and phase transition time proved to be statistically significant, which seems to confirm the correctness, validity, and usefulness of the results.

The melting point is an important factor, especially in the context of applications where phase changes are crucial. Careful choice of components allows for customization of properties to suit a particular application. However, the viscosity of a substance plays a crucial role in various applications, including transportation, manufacturing, and other branches of the industry.

The choice of a specific formulation can be adjusted depending on the viscosity expectations for a given application, and mixtures of different substances can lead to the desired viscosity tailored to specific application needs.

The Pearson correlation coefficient of 0.82 indicates a robust positive association between the average dynamic viscosity and the heat of fusion of the substances examined. This implies that changes in the heat of fusion significantly impact alterations in the dynamic viscosity. Specifically, an increase in the heat of fusion corresponds to a noticeable elevation in the dynamic viscosity, while the composition of the tested substances also has a direct impact on the viscosity. In simpler terms, the examined substances with higher specific heat values tend to have a higher average dynamic viscosity.

This correlation highlights the interconnectedness of these critical properties—heat of fusion and dynamic viscosity. These findings are vital for comprehending material behavior, particularly in the realm of PCMs, where the capacity to store and release thermal energy is crucial. The strong positive correlation suggests that the heat of fusion plays a central role in determining the dynamic viscosity of these substances. This understanding holds importance across various applications, particularly in industries where precise control of material properties is indispensable for optimal functionality.

## Figures and Tables

**Figure 1 materials-17-01538-f001:**
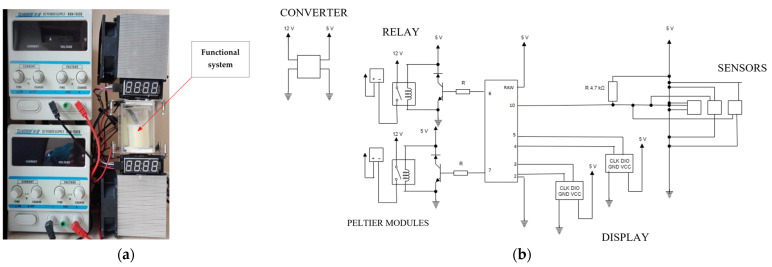
The stand for testing: for measuring the phase change: (**a**) Construction of the test stand and (**b**) an electrical diagram of the test stand.

**Figure 2 materials-17-01538-f002:**
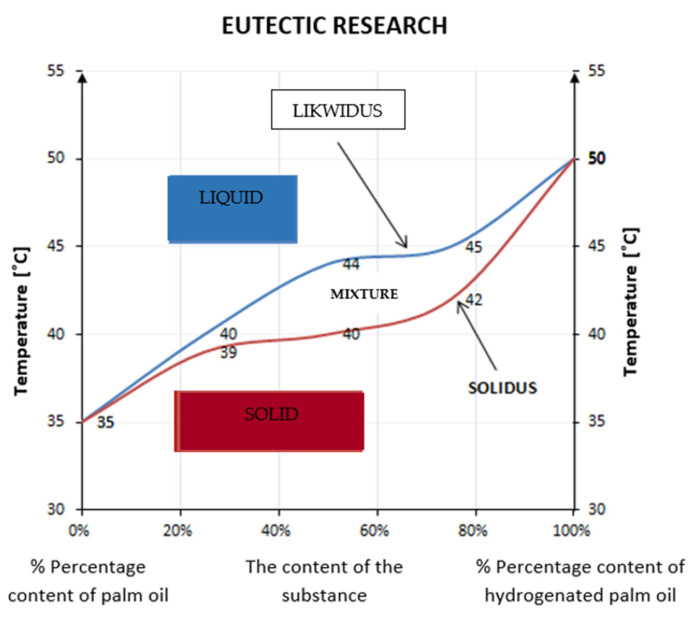
Eutectic of palm oil and hydrogenated palm oil.

**Figure 3 materials-17-01538-f003:**
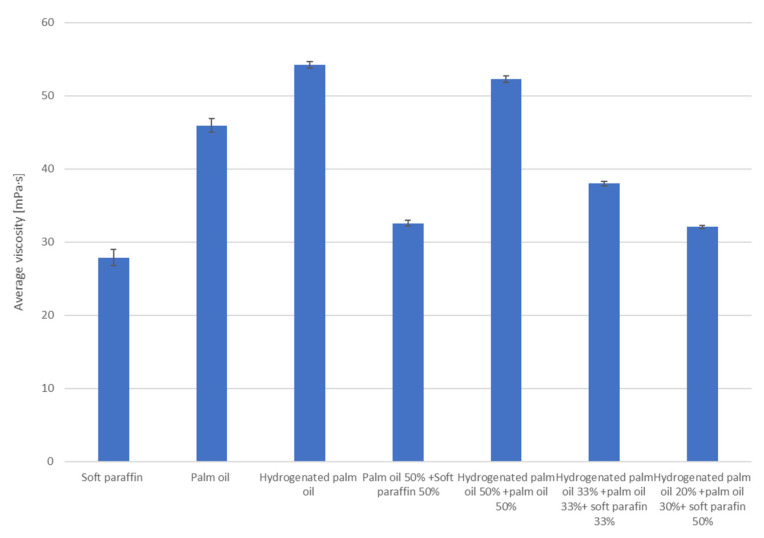
Dynamic viscosity charts for the tested substances with standard deviation [40].

**Figure 4 materials-17-01538-f004:**
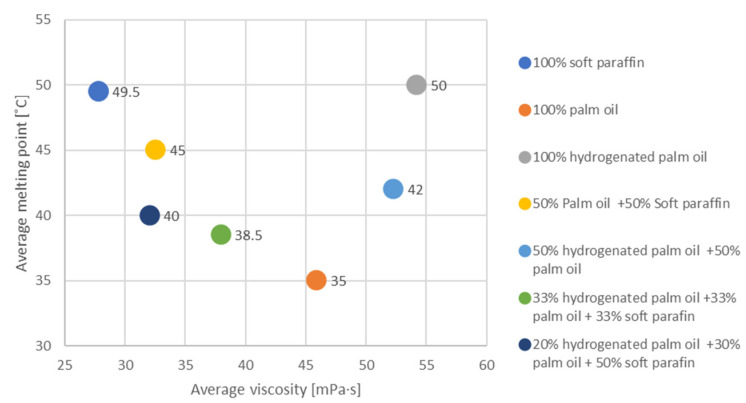
Correlation of melting point and dynamic viscosity at 58 °C.

**Figure 5 materials-17-01538-f005:**
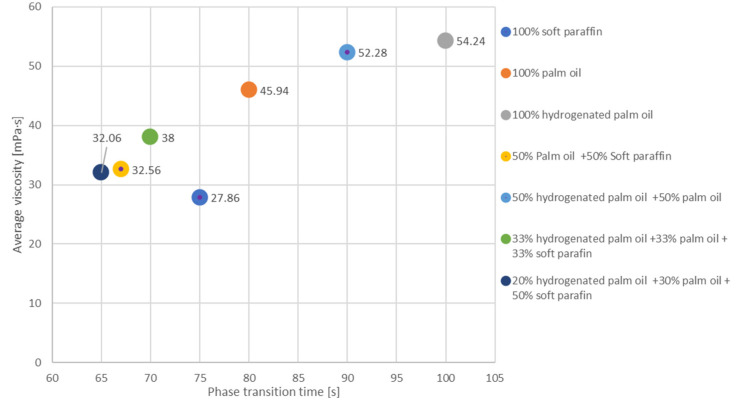
Correlation of phase transition time and dynamic viscosity at 58 °C.

**Figure 6 materials-17-01538-f006:**
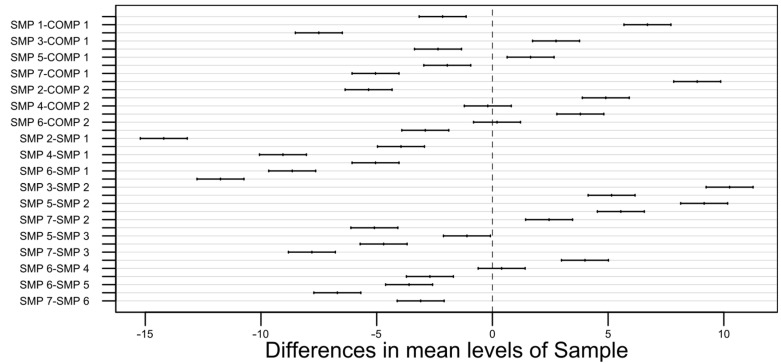
A pair-wise 99% confidence level plot for the minimum transition temperature.

**Figure 7 materials-17-01538-f007:**
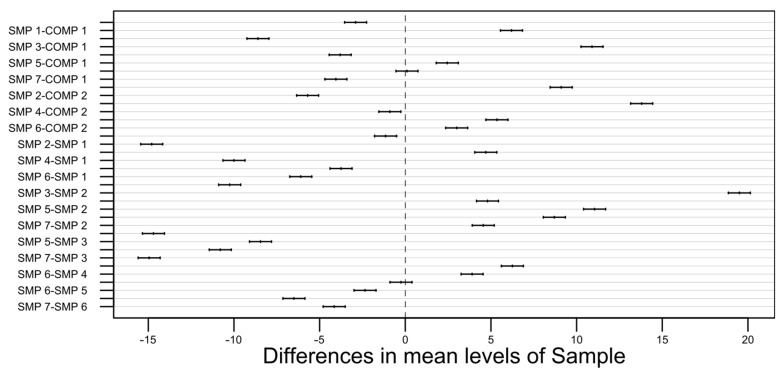
A pair-wise 99% confidence level plot for the maximum transition temperature.

**Figure 8 materials-17-01538-f008:**
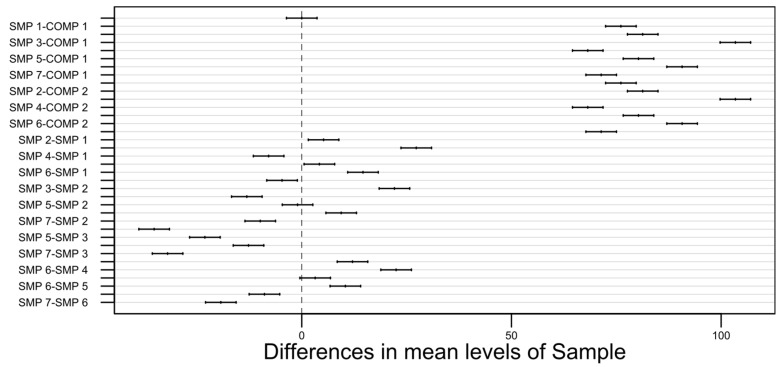
A pair-wise 99% confidence level plot for the phase transition time.

**Figure 9 materials-17-01538-f009:**
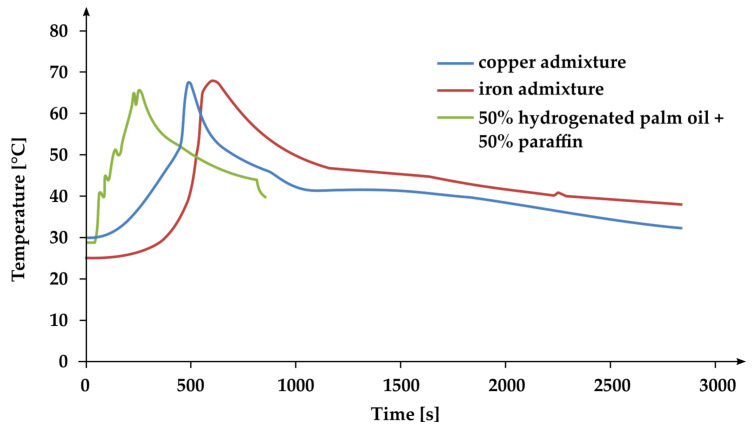
Comparison temperatures and transitions of mixtures 50% hydrogenated palm oil + 50% paraffin with and without the addition of a thermal conductor.

**Figure 10 materials-17-01538-f010:**
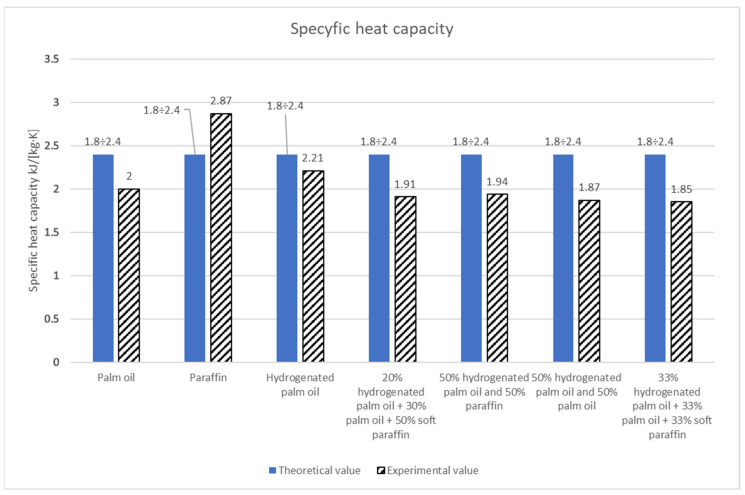
Comparison between theoretical and experimental values of specific heat capacity.

**Figure 11 materials-17-01538-f011:**
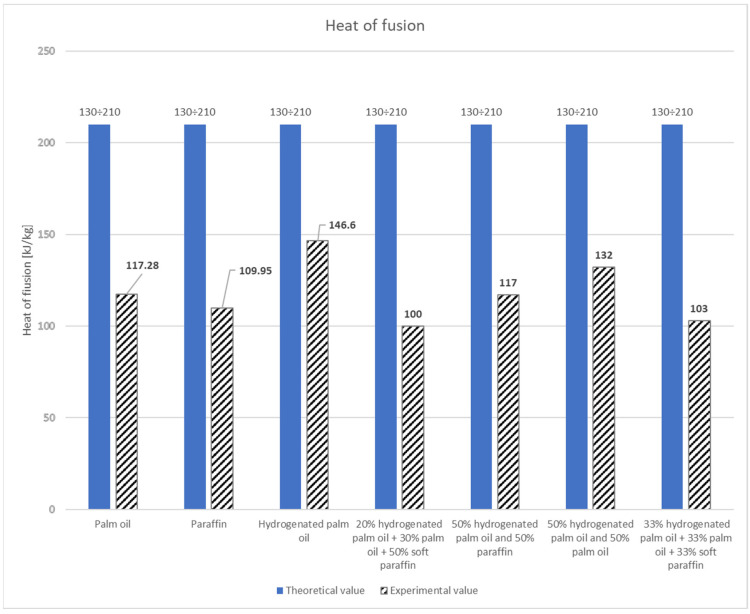
Comparison between theoretical and experimental values of heat of fusion [55].

**Table 1 materials-17-01538-t001:** Examples of PCMs and their properties [13,14].

PCM Name	Type of Material	Process	Phase Change Enthalpy [kJ/kg]	Melting Point [°C]
Paraffin C17	organic	solid–liquid	213	22
Paraffin RT27	organic		179	28
Microencapsulated paraffin (Micronal DS 5001 X)	organic		245	26
Glycerin	organic		199	18
CH_3_(CH_2_)_12_COO(CH_2)2_CH_3_	organic		186	19
CH_3_(CH_2_)_11_OH	organic		189	16–23
Na_2_HPO_4_12H_2_O	inorganic		265	35
Na_2_CO_3_10H_2_O	inorganic		247	34
LiNO_3_·3H_2_O	inorganic		196	30
Mn(NO_3_)_2_·6H_2_O	inorganic		126	26

**Table 2 materials-17-01538-t002:** Properties of palm oil.

Name	Value
Color	White/Yellow
Water content	≤0.1 wt.%
Melting temperature	33–39 °C
Density	0.9 [g/cm^3^] (at 50 °C)
Viscosity	35 [mPa·s]
Boiling point	≥350 °C

**Table 3 materials-17-01538-t003:** Properties of hydrogenated palm oil.

Name	Value
Color	Yellow
Water content	≤0.1 wt.%
Melting temperature	55–58 °C
Density	0.92–0.98 [g/cm^3^] (at 20 °C)
Viscosity	50 [mPa·s]
Boiling point	≥400 °C

**Table 4 materials-17-01538-t004:** Properties of soft paraffin.

Name	Value
Color	White
Water content	≤0.1 wt.%
Melting temperature	45–50 °C
Density	0.755 [g/cm^3^] (at 100 °C)
Viscosity	49.5 [mPa·s]
Boiling point	≥300 °C

**Table 5 materials-17-01538-t005:** Composition of the tested samples.

Number	Palm Oil [wt.%]	Hydrogenated Palm Oil [wt.%]	Soft Paraffin [wt.%]	Additional Material
Sample 1	—	—	100	—
Sample 2	100	—	—	—
Sample 3	—	100	—	—
Sample 4	30	20	50	—
Sample 5	—	50	50	—
Sample 6	50	50	—	—
Sample 7	33	34	33	—
Mixture 1	50	50	—	Copper fines
Mixture 2	50	50	—	Iron fines

**Table 6 materials-17-01538-t006:** Results obtained for the tested samples.

Name	Transition Temperature [°C]	Phase Transition Time [s]	Process Time [s]
100% paraffin	49–50	75	800
100% palm oil	35	80	1200
100% hydrogenated palm oil	45–55	100	800
20% hydrogenated palm oil + 30% palm oil + 50% soft paraffin	40	68	1000
50% hydrogenated palm oil + 50% paraffin	44–46	80	1400
50% hydrogenated palm oil + 50% palm oil	40–44	90	850
33% hydrogenated palm oil + 33% palm oil + 33% soft paraffin sample	37–40	70	1200
50% hydrogenated palm oil + 50% paraffin + copper	42–44	no border	2500
50% hydrogenated palm oil + 50% paraffin + iron	40–41	no border	2700

**Table 7 materials-17-01538-t007:** A one-way ANOVA test for the different substances and dependent variables (DOFs—the number of degrees of freedom, SS—the sum of squares, MSS—the mean of the sum of squares, F-value—test statistic from the F-test, *p*-value—*p*-value of the F-statistic).

Category	DOFs	SS	MSS	F-Value	*p*-Value
Response for minimum transition temperature					
Sample Type	8	2832.1	354.01	393.89	<2.2 × 10^−16^
Residuals	81	72.8	0.90	—	—
Response for maximum transition temperature					
Sample Type	8	2764.65	345.58	2337.5	<2.2 × 10^−16^
Residuals	81	11.97	0.15	—	—
Response for phase transition time					
Sample Type	8	112,387	14,048.4	2951	<2.2 × 10^−16^
Residuals	81	386	4.8	—	—

**Table 8 materials-17-01538-t008:** Comparison of specific heat capacity and dynamic viscosity properties for three base substances [56,57,58].

Substance	Theoretical Heat of Fusion [kJ/kg]	Theoretical Specific Heat Capacity $\left[\frac{kJ}{kg\cdot K}\right]$	Heat of Fusion [kJ/kg]	Specific Heat Capacity $\left[\frac{kJ}{kg\cdot K}\right]$	Dynamic Viscosity [mPa·s]
100% paraffin	130–210	1.8–2.4	109.95	2.87	27.9
100% palm oil			117.28	2	45.9
100% hydrogenated palm oil			146.6	2.21	54.2
20% hydrogenated palm oil + 30% palm oil + 50% soft paraffin			99.7	1.91	32
50% hydrogenated palm oil + 50% paraffin			117.28	1.94	32.56
50% hydrogenated palm oil + 50% palm oil			131.94	1.87	52.28
33% hydrogenated palm oil + 33% palm oil + 33% soft paraffin			102.62	1.85	38

## Data Availability

Data are contained within the article.

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
