# Peer review of "A Study of the Relationship between the Dynamic Viscosity and Thermodynamic Properties of Palm Oil, Hydrogenated Palm Oil, Paraffin, and Their Mixtures Enhanced with Copper and Iron Fines"

_materials, 2024, doi:10.3390/ma17071538_

Round 1

Reviewer 1 Report

Comments and Suggestions for Authors

The manuscript titled "A study of the relationship between dynamic viscosity and thermodynamic properties of Palm Oil, Hydrogenated Palm Oil, Paraffin, and Their Mixtures Enhanced with Copper and Iron Fines" describes some measured thermophysical properties of phase change materials along with enhancing particles. The paper includes measurement data of viscosity, melt temperature, heat of fusion, specific heat, and the time of the phase transformation. I see no issues with how these measurements were made, but I don't think the summary of the results is not sufficient for publication. The conclusions suggest a relationship between the viscosity and the heat of fusion. The results show some relation, but a relation between these properties is not significant or helpful. Also, the paper mentions increasing the thermal conductivity with the enhancing particles, but no measurement of thermal conductivity is presented. I don't feel there is a coherent story in this paper and the work/results presented do not match the need that is articulated in the introduction section. 

Comments on the Quality of English Language

It could be improved, but it is understandable. 

Author Response

Thank you for your evaluation. We have edited the manuscript in response to the three other reviews and changed the outline of the work. Several new conclusions have also been formulated that may be interesting. We believe that the topic of PCM warehouses is very important, especially for materials that are cheap and non-toxic. In some cases, lower viscosity may be more advantageous, allowing faster heat transfer and faster phase transformations, which is what soft paraffins do in our research. However, in other cases, a higher viscosity may be preferred, providing material stability and control over its flow and thermal behavior, examples being substances such as hydrogened palm oil or 50% hydrogened palm oil and 50% palm oil. Among the investigated substances, the highest specific heat is for paraffin. However, the highest heat of transformation was recorded for hydrogenated palm oil, which can be desirable in some PCM applications. Higher specific heat means a greater amount of energy that the PCM can store per unit mass. Higher heat of transformation means a greater amount of energy required for the phase change of the PCM. Such parameters can be advantageous in applications where storing a larger amount of energy is crucial. Despite the popularity of the substances used, there are no similar studies in the literature, which may have a significant impact. We hope that the changes we have made meet your expectations and improve the overall quality of the paper.

Reviewer 2 Report

Comments and Suggestions for Authors

The manuscript deals with phase change materials (PCMs) obtained by combining: palm oil, hydrogenated palm oil and paraffin in different ratios. A new design of a station for PCM testing, based on temperature measurements, has been developed and used. The viscosity of different formulations has been measured and its impact on the heat accumulation has been assessed. In particular, key thermodynamic parameters like: transition time, transition heat, specific heat, etc. have been determined and the specific heat has been found as directly proportional to the average dynamic viscosity.

The manuscript provides useful technical information about the studied PCM systems; for example, the dependence of phase transition time on the average viscosity of the molten PCM. In particular, Authors have found a linear correlation between phase transition time and viscosity and this information is relevant in the PCM selection. However, the manuscript needs to be revised in some of its parts before publication.

- The Introduction Section is not well organized and it must be improved. A short description for the most performant PCMs should be provided. The main technological solutions adopted for solving the problem of a low thermal conductivity should be described. Traditional design of stations for PCMs testing should be presented and a comparison with the innovative design proposed by the Authors should be shortly discussed. References on the above points must be provided. In addition, English is of low quality and the adopted terminology frequently is inadequate.

- Owing to the settling phenomenon, both copper rods (63 microns) and iron powder (150 micron) cannot be used for improving the PCM thermal conductivity. Metals have quite high density values and therefore they promptly settle even in a viscous liquid. In addition, thermal conductivity measurements on such PCM formulations have not been done. As a consequence, I would suggest to remove these only partial results from the manuscript.

- Some clear indication about the way to select the appropriate formulation for tailoring viscosity and therefore the phase transformation time should be provided in the manuscript, since it represents the most relevant of findings of this study.

- The section related to the statistical analysis of the experimental results seems too extended for this type of study and it should be reduced.

Comments on the Quality of English Language

English is of quite low quality and the adopted terminology frequently is not adequate.

Author Response

Dear Reviewer,

Thank you very much for taking the time to carefully read our manuscript. We have accurately read all the comments and referred to all of them. They helped us to significantly improve the article. We have corrected the mistakes, and we hope that now it will meet the standards and receive your recommendations for publication. Below are the general responses to your comments.

Remark 1

The Introduction Section is not well organized and it must be improved. A short description for the most performant PCMs should be provided. The main technological solutions adopted for solving the problem of a low thermal conductivity should be described. Traditional design of stations for PCMs testing should be presented and a comparison with the innovative design proposed by the Authors should be shortly discussed. References on the above points must be provided. In addition, English is of low quality and the adopted terminology frequently is inadequate.

Answer: Thank you for all your comments. We have re-edited the entire introduction and added missing elements.

Remark 2

Owing to the settling phenomenon, both copper rods (63 microns) and iron powder (150 micron) cannot be used for improving the PCM thermal conductivity. Metals have quite high density values and therefore they promptly settle even in a viscous liquid. In addition, thermal conductivity measurements on such PCM formulations have not been done. As a consequence, I would suggest to remove these only partial results from the manuscript.

Answer: The use of falling dispersion is also applicable in the research because we use heating/cooling from specific heat sources, such as Peltier modules. This aspect of the article is just a curiosity that is worth exploring in further research.

Remark 3

Some clear indication about the way to select the appropriate formulation for tailoring viscosity and therefore the phase transformation time should be provided in the manuscript, since it represents the most relevant of findings of this study.

Answer: Thank you for your comment. Added conclusions, e.g.: Among the investigated substances, the highest specific heat is for paraffin. However, the highest heat of transformation was recorded for hydrogenated palm oil, which can be desirable in some PCM applications. Higher specific heat means a greater amount of energy that the PCM material can store per unit mass. Higher heat of transformation means a greater amount of energy required for the phase change of the PCM material. Such parameters can be advantageous in applications where storing a larger amount of energy is crucial.

Remark 4

The section related to the statistical analysis of the experimental results seems too extended for this type of study and it should be reduced.

Answer: Thank you for your comment. An important element is the statistics thanks to which we have confirmation of the repeatability of processes, which is important for heat storage and proves repeatability and stability. We can remove the charts if they are what you think is unnecessary.

Reviewer 3 Report

Comments and Suggestions for Authors

This manuscript studied the relationship dynamic viscosity and thermal dynamic properties of phase change materials. Interesting paper. However, mu comments are as follows,   

 1. Line 55 you can use PCMs instead of Phase-change materials, and hereafter.

2. Line 136, Please leave a space between values and unit.

3. PCMs, PCM, mPCMS, Please unify them.

4. Is Water content ≤ 0.1% by weight?

5. PCM material remove material

6. The percentage of the material is by volume or weight on table 4? Please specify it. Using wt.% if by weight.

7. Please explain what is the parameters on all equations.

8. Suggest using the real content of the material instead of sample 1, 2, 3, etc., Easier for reader to compare.

9. Comparison of what property on Figure 9

Author Response

Dear Reviewer,

Thank you very much for taking the time to carefully read our manuscript. We have accurately read all the comments and referred to all of them. They helped us to significantly improve the article. We have corrected the mistakes, and we hope that now it will meet the standards and receive your recommendations for publication. Below are the general responses to your comments.

Remark 1

Line 55 you can use PCMs instead of Phase-change materials, and hereafter.

Answer: We have left the complete expression phase-change materials only in the abstract, keywords, and the first sentence of the main text.

Remark 2

Line 136, Please leave a space between values and unit.

Answer: It seems to us that you were referring specifically to the degrees Celsius symbol. Therefore, we have put a space between the value and the °C symbol in all the instances.

Remark 3

PCMs, PCM, mPCMS, Please unify them.

Answer: Depending on context, we use either PCM (when referring to a single material or as a modifier in expressions like PCM substance) or PCMs (when referring to several such substances or to phase-change materials in general). The abbreviation mPCMS is shorthand for microencapsulated phase-change material slurry, which was mentioned in the Introduction section; therefore, we cannot change it, because it refers to something else.

Remark 4

Is Water content ≤ 0.1% by weight?

Answer: Yes, it is by weight. We have replaced the percentage sign with the more explicit wt.%, as you suggested in remark no 6.

Remark 5

PCM material remove material

Answer: Thank you for pointing that out. We have now removed the word material, which indeed was redundant in those cases.

Remark 6

The percentage of the material is by volume or weight on table 4? Please specify it. Using wt.% if by weight.

Answer: It is by weight. To remove all ambiguity, we have now explicitly stated that in the table header.

Remark 7

Please explain what is the parameters on all equations.

Answer: The explanation of the parameters has now been added.

Remark 8

Suggest using the real content of the material instead of sample 1, 2, 3, etc., Easier for reader to compare.

Answer: Thank you for pointing that out. We have added detailed descriptions to all charts and descriptions.

Remark 9

Comparison of what property on Figure 9

Answer: Thank you. We missed this. It is the correlation between temperature and the transition. It has been included.

Reviewer 4 Report

Comments and Suggestions for Authors

The authors in the present manuscript show that the phase transition studies in which the following substances were tested and the mixture of: 100% palm oil, 100% paraffin, 100% hydrogenated palm oil, 50% palm oil and 50% paraffin, 50% hydrogenated palm oil and 50% palm oil, 33% hydrogenated palm oil + 33% palm oil + 33% soft paraffin, 20% hydrogenated palm oil + 30% palm oil + 50% soft paraffin, 50% hydrogenated palm oil + 50% palm oil + copper, and 50% hydrogenated palm oil + 50% palm oil + iron . The measurements were carried out on a station for testing phase-change materials, designed specifically for the analysis of phase changes. Viscosity values were also determined for the tested materials, and their potential impact on heat accumulation was assessed. The primary goal of the experiment was to determine some key of thermodynamic parameters, including transition time, transition heat, specific heat, and dynamic viscosity at 58°C. A one-way ANOVA test confirmed the statistical significance of minimum transition temperature, maximum transition temperature, and phase transition time, validating the reliability and utility of the results. The authors should address the following issues and information’s before publication acceptance in the prestigious ‘Materials’ Journal:

1. In Introduction, authors should add a Table that compares the phase-change materials, preparation methods, phase types, and properties with published literatures.

2. In Introduction, authors should explain a bit more about the novelty and importance of this study?

3. In Test stand configuration, how do authors control the chamber temperature (60-70°C)?

4. In Figure 3, authors should explain what is the relationship between viscosity and phase transformation of substances?

5. In Figure 5, can authors incorporate the information for different percentages (10, 20, 30 or 40%) of Palm oil with other?

6. In Introduction section, authors should bit discuss about hydrogen storage for better understanding of energy storage in first paragraph. Authors may go through this publication for more details and cite accordingly: https://doi.org/10.1016/j.micromeso.2017.08.047   

Comments on the Quality of English Language

Minor editing of English language required.

Author Response

Dear Reviewer,

Thank you very much for taking the time to carefully read our manuscript. We have accurately read all the comments and referred to all of them. They helped us to significantly improve the article. We have corrected the mistakes, and we hope that now it will meet the standards and receive your recommendations for publication. Below are the general responses to your comments.

Remark 1

In Introduction, authors should add a Table that compares the phase-change materials, preparation methods, phase types, and properties with published literatures.

Answer: Thank you for your attention. The possible types of PCMs are initially presented. An example table containing thermodynamic parameters has also been added.

Remark 2

In Introduction, authors should explain a bit more about the novelty and importance of this study?

Answer: Thank you; good point. Despite the widespread use of the materials used for testing, there are not many studies showing, for example, properties for mixtures. The  justification is included in the introduction

Remark 3

In Test stand configuration, how do authors control the chamber temperature (60-70°C)?

Answer: Thank you for pointing out this part in the text; this information was missing. It has been added. The research station has sensors for measuring temperature on the upper and lower Peltier modules and in the beaker with the material. The temperature of the modules was controlled via the program in the range of 60-70 °C.

Remark 4

In Figure 3, authors should explain what is the relationship between viscosity and phase transformation of substances?

Answer: Thank you; good point. Thanks to the viscosity test, it is possible to determine which substance exhibits the most favorable flow characteristics. Consequently, among the tested substances, it is possible to identify the one in which heat transfer will occur most effectively.

Remark 5

In Figure 5, can authors incorporate the information for different percentages (10, 20, 30 or 40%) of Palm oil with other?

Answer: Thank you for drawing our attention to this aspect. The research is currently closed but could be continued it in the future.

Remark 6

In Introduction section, authors should bit discuss about hydrogen storage for better understanding of energy storage in first paragraph. Authors may go through this publication for more details and cite accordingly: https://doi.org/10.1016/j.micromeso.2017.08.047

Answer: Thank you for your comment. Similarities between different types of storage have been indicated.

Round 2

Reviewer 2 Report

Comments and Suggestions for Authors

Authors have carefully revised their manuscript and answered all questions. Therefore, the manuscript can be accepted in its present form.

Comments on the Quality of English Language

English is fine only some minor editing is required.